# Impact of household size and co-resident multimorbidity on unplanned hospitalisation and transition to care home

Clare MacRae [1,2,3] ✉, Stewart W. Mercer[1,2], Eleojo Abubakar[4], Andrew Lawson [2,5], Nazir Lone [1], Anna Rawlings[6], Jane Lyons[6], Ronan A. Lyons [6], Amy Mizen[6], Rich Fry[6], Gergő Baranyi[7], Jamie Pearce[8], Chris Dibben[8], Karin Modig[3], Rhiannon Owen [6] & Bruce Guthrie [1,2]

The ability to manage ill health and care needs might be affected by who a person lives with. This study examined how the risk of unplanned hospitalisation and transition to living in a care home varied according to household size and co-resident multimorbidity. Here we show results from a cohort study using Welsh nationwide linked healthcare and census data, that employed multilevel multistate models to account for the competing risk of death and clustering within households. The highest rates of unplanned hospitalisation and care home transition were in those living alone. Event rates were lower in all shared households and lowest when co-residents did not have multimorbidity. These differences were more substantial for care home transition. Therefore, living alone or with co-residents with multimorbidity poses additional risk for unplanned hospitalisation and care home transition beyond an individual's sociodemographic and health characteristics. Understanding the mechanisms behind these associations is necessary to inform targeted intervention strategies.

The increasing prevalence of multimorbidity - the co-existence of multiple long-term conditions - is driven primarily by an ageing population and improved survival from acute illness[1]. Multimorbidity presents a growing challenge because it is associated with more frequent use of health and social care services[2,3]. While multimorbidity in an individual is known to be strongly associated with increased unplanned hospitalisation[1] and transitioning to live in a care home[4], the influence of household size and composition, particularly the multimorbidity status of household co-residents, is less well understood[5].

People with multimorbidity often depend on family members to care for and support them in managing their conditions[6]. The households in which people live plausibly influence whether someone uses unscheduled healthcare or can continue to live in their own home. Household co-residents can provide instrumental support, such as direct physical care or informational support, such as interpreting written material required to negotiate complex healthcare systems[7]. Membership of a household with one or more co-residents can also improve social integration, support, and social ties[7]. In contrast to this, living alone can be associated with isolation and increased use of unplanned hospital care[8], and it is a growing concern as it becomes more common. For example, the number of people living alone in the UK increased by 8% between 2013 and 2023[9]. However, even when

[1]Advanced Care Research Centre, University of Edinburgh, Bio Cube 1, Edinburgh BioQuarter, 5 Little France Rd, Edinburgh EH16 4UX, UK. [2]Usher Institute, College of Medicine and Veterinary Medicine, University of Edinburgh, 5 Little France Rd, Edinburgh EH16 4UX, UK. [3]Unit of Epidemiology, Institute of Environmental Medicine, Karolinska Institutet, Stockholm, Sweden. [4]School of Social and Political Science, University of Edinburgh, Chrystal Macmillan Building, Edinburgh, UK. [5]Medical University of South Carolina, Department of Public Health Sciences, Charleston, SC, USA. [6]Swansea University Medical School, Data Science Building, Singleton Campus, Swansea, UK. [7]Centre for Longitudinal Studies, UCL Institute of Education, University College London, London, United Kingdom. [8]University of Edinburgh Institute of Geography, Institute of Geography Edinburgh, Edinburgh, UK. ✉e-mail: clare.macrae@ed.ac.uk

someone lives with other people, if their co-resident has multi-morbidity, then the potential for household members to support them may be reduced because of the lower capacity of their co-resident to provide help to manage chronic illness and/or reduced resilience to acute events. Additionally, providing care for a co-resident with multimorbidity might affect health and predispose the individual to higher health and social care use. Several studies indicate that informal caregiving responsibilities can adversely affect the caregiver's physical and mental health[10], and this is particularly the case where the care recipient has complex health issues[11]. These existing studies, however, characterise the effect of caring responsibilities using self-reported health information gathered by retrospective recall. This might limit representativeness because people living with those affected by the most complex health needs might not be included in surveys because of the substantial demands that their caring responsibilities place on their time.

Unplanned hospitalisation and transition to living in a care home are events which most people wish to avoid if possible and are associated with individual characteristics, such as age, sex, and number of long-term conditions[12]. However, it is unclear whether characteristics such as household size and co-resident multimorbidity of different kinds (e.g., physical-mental multimorbidity[13]) are independently associated with these outcomes. Using Census-linked population data from Wales, this analysis aimed to explore whether household size and co-resident multimorbidity were associated with unplanned hospitalisation and transition to living in a care home.

## Results

A total of 1,472,185 community-dwelling adults aged 18 years and over on 27 March 2011 were included in the study. The mean age of the study population was 50.6 years, 778,574 (52.9%) were women, and the mean number of long-term conditions was 1.6. People living alone comprised 299,698 (20.4%) of participants and 35.4% of households, and those living alone were, on average, the oldest (mean age 59.5 years), included a higher proportion of women (57.1%), and had the highest mean number of conditions (2.4) of the study population. The 617,395 (41.9%) of participants living in the 252,958 (29.9%) households with three or more residents were, on average, the youngest (mean age 40.6 years), had the lowest proportion of women (51.0%), and the lowest mean number of conditions (0.8) (Table 1). During follow-up to 26 March 2016, 386,327 (26.2%) of people had at least one unplanned hospital admission, 14,217 (1.0%) transitioned to live in a care home, 89,023 (6.0%) died, and 146,093 (9.9%) migrated to live outside Wales.

Multimorbidity was present in 538,762 (36.6%) of the whole study cohort (Table 2). Multimorbidity was relatively more prevalent in smaller households. 29.6% of people with multimorbidity lived in single-person households compared to 20.4% of the total study population, and 47.1% of people with multimorbidity lived in two-person households compared to 37.7% of the total study population (Table 2). In contrast, only 23.3% of people with multimorbidity lived in three-or-more-person households compared to 41.9% of the study population.

People living alone had the highest rate of unplanned hospitalisation (91.96/1000 person-years), with lower rates in all other household arrangements (ranging from 39.88/1000 person-years in three-person households where co-residents did not have multimorbidity to 85.46/1000 person-years in two-person households where the co-resident did have multimorbidity) (Table 3).

Effect sizes and the differences in hazard ratios between living arrangements were the largest in the unadjusted models. The greatest attenuation was seen when age, sex, and socioeconomic deprivation were incorporated (in the partially adjusted models). A smaller degree of attenuation was seen with the addition of the number of long-term conditions, body mass index, and smoking in the fully adjusted models (Table 3, Fig. 1, Supplementary Table S9). For example, in those living in three-or-more-person households, compared to living alone, the unadjusted HR of unplanned hospitalisation when no co-residents had multimorbidity was lowest (aHR 0.37, 95% CI 0.37–0.37) and was intermediate when at least one co-resident had multimorbidity (aHR

**Table 1 | Characteristics of the study population according to household siz**

| Characteristic | Whole study population | Single person households | Two person households | Three-or-more person households |
|---|---|---|---|---|
| Number of individuals | 1,472,185 (100) | 299,698 (20.4) | 555,092 (37.7) | 617,395 (41.9) |
| Number of households | 845,182 (100) | 299,698 (35.4) | 295,526 (35.0) | 252,958 (29.9) |
| Age, years | | | | |
| Mean (SD) | 50.6 (18.7) | 59.5 (19.7) | 56.8 (17.2) | 40.6 (14.4) |
| 18–30 | 248,180 (16.9) | 29,785 (9.9) | 59,775 (10.8) | 158,620 (25.7) |
| 30–45 | 360,159 (24.5) | 47,193 (15.7) | 77,600 (14.0) | 235,366 (38.1) |
| 46–65 | 510,973 (34.7) | 93,180 (31.1) | 225,105 (40.6) | 192,688 (31.2) |
| 66–85 | 308,201 (20.9) | 101,365 (33.8) | 179,043 (32.3) | 27,793 (4.5) |
| 86+ | 44,672 (3.0) | 28,175 (9.4) | 13,569 (2.4) | 2,928 (0.5) |
| Sex | | | | |
| Men | 693,611 (47.1) | 128,472 (42.9) | 262,355 (47.3) | 302,784 (49.0) |
| Women | 778,574 (52.9) | 171,226 (57.1) | 292,737 (52.7) | 314,611 (51.0) |
| Socioeconomic position | | | | |
| 1 (lowest) | 275057 (18.7) | 64984 (21.7) | 92550 (16.7) | 117523 (19.0) |
| 2 | 300894 (20.4) | 64645 (21.6) | 110216 (19.9) | 126033 (20.4) |
| 3 | 303544 (20.6) | 61154 (20.4) | 116442 (21.0) | 125948 (20.4) |
| 4 | 274284 (18.6) | 52853 (17.6) | 109265 (19.7) | 112166 (18.2) |
| 5 (highest) | 318406 (21.6) | 56062 (18.7) | 126619 (22.8) | 135725 (22.0) |
| Long-term conditions | | | | |
| Mean number (SD) | 1.6 (2.0) | 2.4 (2.5) | 1.9 (2.1) | 0.8 (1.4) |
| Multimorbidity: ≥2 long-term conditions | 538,762 (36.6) | 159,205 (53.1) | 253,970 (45.7) | 125,587 (20.3) |

**Table 2 | Distribution of multimorbidity and mental-physical multimorbidity across households of different sizes**

| Households in the total study population N (of households) = 845,182 | Individuals with multimorbidity N (of individuals) = 538,762 Number (%) of individuals | Individuals in the total study population N (of individuals) = 1,472,185 Number (%) of individuals |
|---|---|---|
| Single person households N (of households) = 299,698 | 159,205 (29.6) | 299,698 (20.4) |
| Two person households N (of households) = 295,526 | 253,970 (47.1) | 555,092 (37.7) |
| Three-or-more person households N (of households) = 252,958 | 125,587 (23.3) | 617,395 (41.9) |

**Table 3 | Associations between household size and co-resident multimorbidity with first unplanned hospitalisation and transitioning to live in a care home; event rate per 1000 person-years and hazard ratios (HRs) with 95% confidence intervals (95% CIs) for unadjusted and fully adjusted models**

| Covariate | Event rate/1000 person-years | Unadjusted HR (95% CI) | Fully adjusted HR (95% CI)[a,b] |
|---|---|---|---|
| Outcome: Unplanned hospitalisation | | | |
| Lives alone | 91.96 (91.41–92.51) | Reference (1.00) | Reference (1.00) |
| 2-person household co-resident multimorbidity | 85.46 (84.89–86.04) | 0.88 (0.88–0.89) | 0.91 (0.90–0.92) |
| 2-person household co-resident no multimorbidity | 53.85 (53.45–54.24) | 0.55 (0.54–0.55) | 0.89 (0.89–0.90) |
| 3-person household co-resident multimorbidity | 53.12 (52.63–53.62) | 0.50 (0.50–0.51) | 0.92 (0.91–0.93) |
| 3-person household co-resident no multimorbidity | 39.88 (39.6–40.17) | 0.37 (0.37–0.37) | 0.87 (0.86–0.88) |
| Outcome: Transition to live in a care home | | | |
| Lives alone | 5.88 (5.76–6.00) | Reference (1.00) | Reference (1.00) |
| 2-person household co-resident multimorbidity | 2.82 (2.73–2.92) | 0.76 (0.75–0.77) | 0.78 (0.77–0.79) |
| 2-person household co-resident no multimorbidity | 0.70 (0.66–0.74) | 0.30 (0.29–0.30) | 0.71 (0.69–0.72) |
| 3-person household co-resident multimorbidity | 0.73 (0.68–0.78) | 0.23 (0.23–0.24) | 0.78 (0.75–0.80) |
| 3-person household co-resident no multimorbidity | 0.14 (0.12–0.16) | 0.08 (0.08–0.08) | 0.57 (0.55–0.59) |

[a]Fully adjusted model for unplanned hospitalisation incorporates age group, sex, socioeconomic position, number of long-term conditions, smoking, and alcohol.
[b]Fully adjusted model for transition to living in a care home incorporates age group, sex, socioeconomic position, number of long-term conditions, body mass index, and alcohol.

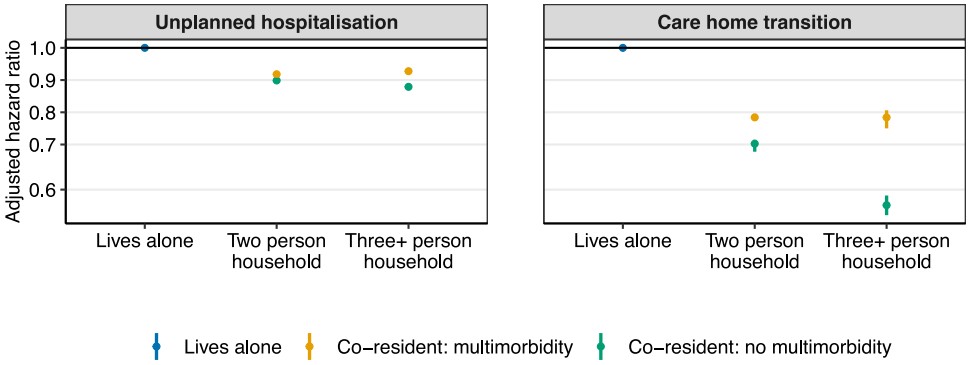

**Fig. 1 | Associations between co-resident multimorbidity and household size with unplanned hospitalisation and transition to living in a care home; adjusted hazard ratios and 95% confidence intervals.** Source data are provided as a Source Data file. The number of people living in each household arrangement was: 299,698 lived alone, 307,317 in two-person households where the co-resident had multimorbidity, 247,775 in two-person households where the co-resident did not have multimorbidity, 423,987 in three-or-more-person households where one or more co-residents had multimorbidity, and 193,408 in three-or-more-person households where no co-residents had multimorbidity.

0.50, 95% CI 0.50–0.51). The same comparison in the fully adjusted models showed a similar pattern but weaker associations: aHR 0.87 (95% CI 0.86–0.88) compared to aHR 0.92 (95% CI 0.91–0.93). This effect was restricted to people living in three-or-more-person households, as there was little difference in aHR for those living in two-person households, regardless of whether their co-resident had multimorbidity (0.91, 95% CI 0.90–0.92) or not (0.89, 95% CI 0.89–0.90).

The rate of transitioning to live in a care home was also highest in people who lived alone (5.88/1000 person-years) and was much lower

in those living in other household arrangements (from 0.14 in three-or-more-person households with a co-resident who did not have multimorbidity to 2.82 in two-person households where the co-resident did have multimorbidity) (Table 3).

Before and after partial and full adjustment for demographic, health, and health-related behaviours, the difference between living alone and the highest-risk group of the co-habiting living arrangements was considerable and was larger than seen in the unplanned hospitalisation models. However, a similar degree of attenuation to the

unplanned hospitalisation models was seen in the magnitude of effect sizes, and the difference in hazards between living arrangements was seen when sociodemographic characteristics were included in the partially adjusted model and health behaviours in the fully adjusted model (Supplementary Table S10).

In people living in three-or-more-person households, compared to living alone, the fully adjusted aHR of transition to care home was 0.57 (95% CI 0.55–0.59) when no co-residents had multimorbidity, which was substantially lower than when one or more co-residents had multimorbidity (0.78, 95% CI 0.75–0.80) (Table 3). The same comparison of the difference in aHRs in two-person households was smaller but still moderate in size. Living with a co-resident who did not have multimorbidity was associated with the lowest hazard (aHR 0.71, 95% CI 0.69–0.72), and there was a significantly higher hazard for those living with a co-resident who did have multimorbidity (0.78, 95% CI 0.77–0.79).

In all subgroup and sensitivity analyses, living alone was associated with the highest HRs of both unplanned hospitalisation and of transitioning to live in a care home. Similarly, in those who lived with other people, a higher hazard of both outcomes was found in those who had co-residents with rather than without multimorbidity.

Living with coresident(s) with versus without multimorbidity was associated with a slightly higher aHR of unplanned hospitalisation in three-or-more-person households in the younger and in two-person households for the older subgroup (Supplementary Table S11). In models examining the transition to living in a care home, there were larger differences in aHR between those who did and did not live with co-residents with multimorbidity in the younger versus the older subgroup (Supplementary Table S12). The difference in the hazard of unplanned hospitalisation and of transitioning to live in a care home for individuals living with co-residents who did or did not have multimorbidity was more marked for men than women (Supplementary Table S13). There were no substantial differences in the hazard of unplanned hospitalisation or transition to living in a care home when defining co-resident multimorbidity as the co-existence of both mental and physical long-term conditions. There were no substantial differences in the hazard of unplanned hospitalisation or transition to living in a care home when defining co-resident multimorbidity as the co-existence of both mental and physical long-term conditions. Finally, there was no difference in the hazard of unplanned hospitalisation or transition to living in a care home between different numbers of co-resident long-term conditions.

## Discussion

In this study, we find that the prevalence of multimorbidity was higher in single and two-person households than in larger households. People who lived alone were substantially more likely than those living with others to experience unplanned hospitalisation or transition to living in a care home. In two-person and larger households, living with co-residents who had (compared to not having) multimorbidity was associated with a marked increase in HR of unplanned hospitalisation and transition to living in a care home in unadjusted analyses. These associations were attenuated after adjustment for individual characteristics but remained statistically significant for both outcomes, although the adjusted strength of association for co-resident multimorbidity was small for unplanned hospitalisation and more substantially different for transitioning to live in a care home.

This study has several strengths. We used a large population dataset with linkage across health, administrative, and census data, which is likely less susceptible to selection bias than consented research studies. The study population was defined from Census data, which provided a robust method of characterising households at baseline. The analyses used conditions recommended for multimorbidity research[14], adjusted for demographic, health, and health-

related behaviours, and included several subgroup and sensitivity analyses.

Our study also has several limitations. The study assessed the multimorbidity status of household co-residents using hospital inpatient and primary care data, which improved ascertainment[15]. However, this does not account for conditions that are undiagnosed or not recorded by hospital or primary care staff, an issue common to all studies using electronic health record data. A total of 11.5% of households were excluded because one or more residents did not have linked primary care data. However, incorporating primary care data considerably improves the robustness of multimorbidity measurement[15] and was therefore prioritised in the study design over including all households but only ascertaining morbidity using hospital inpatient data. Potential confounders of the associations we were unable to measure with the available data include the provision of both formal and informal home care (including from family or friends who were not co-residents) or how co-residents are related to one another, both of which are likely to be important. Finally, this study used data from NHS Wales in the UK, and the results may not be generalisable to countries with different health and social care provision or different patterns of informal care.

Our study finds that living alone was associated with a small increased aHR of unplanned hospitalisation, and studies examining the association between living alone and other health outcomes reported similar findings. For example, a recent study, also from the UK, examined emergency department attendance and general practitioner appointments in 1447 older adults. Both outcomes were more common for people who lived alone compared to those living in larger households (adjusted odds ratio [aOR] 1.50 [95% CI 1.16–1.93] and 1.40 [95% CI 1.04–1.88] respectively)[16]. The observed associations are stronger than in this study, although there were fewer covariates adjusted for emergency department attendance and GP appointments, which are more discretionary events given that a person is less able to control whether or not they are admitted to hospital.

Given that people with multimorbidity often rely on care from family members and informal networks[17], we suggest that the availability or absence of care from co-residents is likely to translate into protection or vulnerability, respectively, to the requirement for long-term care. A scoping review describing the experiences and support needs of informal caregivers of people with multimorbidity identified additional responsibilities and burdens, such as managing multiple medications, uncertainty around understanding and managing the care recipient's symptoms, and lack of support[18]. These challenges can result in increased stress and depression[19], emotional and psychosocial morbidity associated with caring and social isolation resulting from this[18,20], and financial difficulties[20], that result in a negative impact on how they manage their health conditions[21]. Therefore, people living with co-residents with multimorbidity might be at increased risk of becoming overwhelmed in the event of their own illness, which then affects the capacity of the household support network to care for someone who requires additional support. These factors might trigger an episode of unplanned hospitalisation that might not have been necessary in other circumstances. A qualitative study reported that transition to a care home can result when the total care needs within a household become unsustainable, resulting in long-term residential care requirements[22]. This situation might be more likely to occur where a person experiencing increased care needs lives with co-residents with multimorbidity, resulting in earlier and more exaggerated instability in household coping mechanisms because adequate support is not available. However, our results show that this is less important for the requirement of unplanned inpatient care. People living with co-residents with multimorbidity were more likely to experience unplanned hospitalisation in unadjusted analyses, with substantial effect sizes. In contrast, when individual characteristics were accounted for analyses examining unplanned hospitalisation for

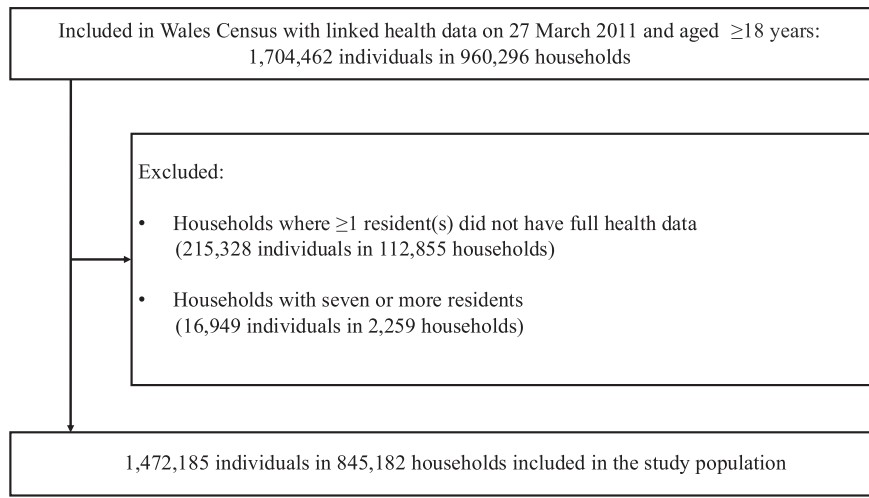

**Fig. 2 |** Wales Census and health data were linked to create the study cohort.

people living alone or with co-residents with multimorbidity, the associations remained statistically significant but with small effect sizes. This suggests that the association is mainly driven by variation in the demographic and health characteristics of individuals living in different households, albeit with some residual effect relating to the presence or absence of a co-resident or co-resident multimorbidity.

Potential pathways linking co-resident multimorbidity status with unplanned hospitalisation might involve social and material factors. For example, the availability of routine healthcare might be more limited when there are conflicting healthcare needs within one household and diminished financial resources that can accompany chronic illness[20]. Therefore, policy measures such as providing co-residents of individuals with multimorbidity with access to care coordinators, support groups, and greater financial[6] and targeted multi-disciplinary support[23] might mitigate some of the increased likelihood. Research and policy responses must consider all contributory factors such as the person, multimorbidity affecting any household resident, available health and care services, and the interrelationship between these factors and the household and wider environment.

It is understood that there is a synergistic effect of experiencing concordant physical and mental conditions that result in a higher level of disability and deterioration in function than those with physical health conditions only[13]. However, our study finds that there were few differences in the effect of living with a co-resident with multi-morbidity compared with mental-physical multimorbidity on unplanned hospitalisation and transition to a care home. Further research is needed to describe other and more granular definitions of multi-morbidity to determine if definitions with a higher threshold or those pertaining to certain body systems could relate to increased risk, for example, where there might be higher associated care needs.

Future research would ideally incorporate measurements of accumulated individual and household exposures over the life course to understand how physical and mental health, material deprivation, household social dynamics including household kinship and care relationships, and health and care services might mutually reinforce each other over time. Using more granular datasets that provide an understanding of the reasons for living alone by life stage and the mechanisms by which living alone or with co-residents with multi-morbidity in different ages and birth cohorts increases the event rate of unplanned hospitalisation and transition to living in a care home could be helpful precursors for the development and economic evaluation of complex interventions.

Our study finds that living alone and living with co-residents with multimorbidity is independently associated with the transition to living in a care home and, to a smaller extent, with unplanned hospitalisation, which is clinically meaningful given that it is a common event. These associations are most likely to represent interacting and reinforcing mechanisms relating to additional care needs, and the challenges faced when navigating fragmented health and care systems. This burden will increase as the population ages, when it will become more common for people to live with co-residents who have multi-morbidity, and multimorbidity and associated care needs are likely to become more complex. Therefore, focused research and public health attention are needed to identify individuals and households who would benefit from targeted support.

## Methods

This research complies with all relevant ethnical regulations and has been reviewed by the SAIL IGRP and RAP panels. In this cohort study, we used data from SAIL Databank, which linked census, health, and administrative databases using a unique personal identification number[24]. The study population was people and households identified in the decennial national Wales Census on 27 March 2011[25]. Eligible individual participants were aged 18 years and over and living in households with one to six people where all of the residents living within the household were registered with a GP practice that contributes data to SAIL Databank (see Fig. 2 for study cohort selection). Individuals were followed up until the first of 26 of March 2016, emi-gration from Wales, or death. Supplementary Table S1 provides an overview of the included datasets, variables, and linkage methods. Identification of households.

Linkage between household co-residents used the Wales Census 2011 unique household identifier, which meant we could accurately identify people living in the same household. All individuals, including child co-residents (age <18 years old) living in households of eligible adult participants, were accounted for in the measurement of house-hold size. Child co-residents were not included in counts of co-resident multimorbidity status, given that the study aimed to ascertain the effect of differing levels of potential care available from household co-residents. Supplementary Table S2 provides summary statistics regarding the distribution of children across households included in the study.

Multimorbidity was measured on 27 March 2011 and was defined as two or more long-term conditions[26] from a list of 47. Methods used to select the constituent long-term conditions (based on a recent Delphi consensus study on the measurement of multimorbidity[14] and a quantitative examination of the implications of considering different numbers and selections of long-term conditions in the count of

multimorbidity[27]) and to define each long-term condition according to each data source[28] are described in Supplementary Information Panel S1. Briefly, primary care data included diagnosis codes, prescribing, and laboratory data; secondary care data included hospital discharge diagnosis codes. Ascertainment of only active conditions was achieved by using varying look-back duration for long-term condition codes and recent prescribing data as described in our previous study using the same dataset[28] (Supplementary Tables S3-4 describe the methods for ascertainment of conditions, and Supplementary Table S5 shows the prevalence of each of the included conditions).

Two outcomes were measured: (1) time to the first unplanned hospitalisation (using the Patient Episode for Wales Dataset), and (2) time to transition to a care home (using linkage of the Welsh Demographic Services Dataset and the SAIL Care Homes dataset). Participants were followed up until the first of either the outcome being studied, death, move from Wales, or the end of the study.

All adjusted models contained information on covariates accounting for the demographic, clinical, and health-related behaviours of individuals, including age group (centred at 46–65 years), sex (as recorded in the electronic health record), ethnicity, socioeconomic position, number of long-term conditions, smoking, alcohol, and body mass index (Supplementary Information Panel S2 and Supplementary Tables S6–8). Socioeconomic position was categorised using deciles of the 2011 Welsh Index of Multiple Deprivation, the Welsh Government's official measure of relative deprivation[29] for every Lower Super Output Area (statistical geographies comprising between 400 and 1,200 households) in Wales[30].

First, we described the study population according to demographic characteristics and the prevalence of multimorbidity in individuals according to the size of their household. Second, we estimated the hazard ratio (HR) and 95% confidence intervals (95% CIs) for unplanned hospitalisation and transition to living in a care home (using separate models for each outcome) for household size and co-resident multimorbidity. To do this, we created an exposure variable of interest by categorising each participant into one of five groups according to their living arrangement: living alone, two-person household where the co-resident had multimorbidity, two-person household where the co-resident did not have multimorbidity, three-or-more-person household where one or more co-residents had multimorbidity, and three-or-more-person household where no co-residents had multimorbidity. In addition to the primary analyses, subgroup and sensitivity analyses were performed. First, to examine how the effect of household size and co-resident multimorbidity might vary for particular segments of the population, given that these exposures might affect people differently depending on their age, subgroup analyses were performed in individuals aged 18-64 and those aged 65 and older. Second, to test whether the effect of co-resident multimorbidity was different between men and women, in case gender roles and cultural expectations predisposed to different effects by sex, and to support the generalisability and translation of research findings, we described associations separately for men and women. Third, household size and co-resident mental-physical multimorbidity - defined as one or more physical health and one or more mental health long-term conditions - were tested in a sensitivity analysis to examine whether the same or different results were found using an alternative definition of multimorbidity[26]. Finally, a more granular examination of co-resident morbidity was assessed by disaggregating the exposure variable of interest, replacing multimorbidity as a binary variable with zero to one, two to three, or four or more long-term conditions.

Multistate Cox proportional hazard models were used to examine the hazard ratio of each outcome using separate models. Coefficients for transition one (transitioning from home to first unplanned hospitalisation or home to living in a care home, for separate models) were reported in the results section. The remaining two transitions, from home to death (therefore allowing the model to incorporate the competing risk of death) and transition three, which was unplanned hospitalisation or care home to death, were not reported. No violations of the proportional hazard assumption were found on visual inspection for deviation from the zero slope of plotted Schoenfeld residuals. The outcome was measured using the appropriate data structure by incorporating household-level random effects to account for the clustering of effects within household units. The Akaike Information Criterion guided variable selection to assess model fit. Model coefficients were presented for unadjusted, partially adjusted (incorporating age group, sex, and socioeconomic position), and fully adjusted models.

Data were extracted from IBM DB2 relational databases using Structured Query Language. Further data manipulation, statistical analyses (using the coxme()function), and plotting were performed using R version 4.1.2.

The project was approved by the SAIL Databank independent Information Governance Review Panel[24] (project number 1350), and the use of linked data from Wales Census 2011 was approved by the Office of National Statistics Research Accreditation Panel (Digital Economy Act project number 2022/139). Written consent from participants was not necessary due to the use of retrospective anonymised electronic health records data. All data was used according to the terms and conditions of the SAIL Databank databases where the data was sourced, and all research outputs were manually screened and cross-checked against the approved research protocol before being released from the secure research environment.

The funders of the study had no role in study design, data collection, data analysis, data interpretation, or writing of the report. The corresponding author had full access to all the data used in the study and had final responsibility for the decision to submit the study for publication.

**Reporting summary**
Further information on research design is available in the Nature Portfolio Reporting Summary linked to this article.

## Data availability

The datasets used in this study are available in anonymised form via a secure data sharing platform, underpinned by the ISO 27001 internationally recognised best practice standard for an Information Security Management System, and compliant with National Research Ethics Service guidance. Access to the data is available only for accredited researchers using a secure remote desktop login and following approval for a project via an application to the SAIL IGRP (https://saildatabank.com/governance/). However, any bona fide researcher can apply to access the same data by applying to the original data holders. All data released from the SAIL Databank safe haven is available within the Supplementary Material and Source Data file. As data from the 2011 Wales Census was used, the project underwent additional approvals by the Research Accreditation Panel (RAP); DEA Accredited Project Number is 2022/139. Source data are provided with this paper.

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

## Acknowledgements

Medical Research Council MR/W000253/1 fellowship for C.M.; National Institute for Health Research (NIHR) Artificial Intelligence and Multimorbidity for B.G. and S.W.M.: Clustering in Individuals, Space and Clinical Context (AIM-CISC) grant NIHR202639; Legal and General plc funding for the Advanced Care Research Centre for B.G. and S.W.M.

## Author contributions

Conceptualisation C.M., B.G.; data curation C.M., B.G.; formal analysis C.M., B.G.; funding acquisition C.M., B.G., S.W.M.; investigation and methodology C.M., B.G., R.K.O., A.L., N.L., K.M.; project administration C.M.; supervision B.G., S.W.M., C.D., N.L.; writing - original draft C.M.; writing - review and editing C.M., B.G., S.W.M., E.A., A.L., N.L., R.K.O., A.R., J.L., R.A.L., A.M., R.F., G.B., J.P., C.D., K.M.

## Competing interests

R.K.O. is a member of the National Institute for Health and Care Excellence (NICE) Technology Appraisal Committee, member of the NICE Decision Support Unit (DSU), and associate member of the NICE Technical Support Unit (TSU). She has served as a paid consultant to the pharmaceutical industry and international reimbursement agencies, providing unrelated methodological advice. She reports teaching fees from the Association of British Pharmaceutical Industry (ABPI). All other authors declare that they have no competing interests.
