## [Transparent Peer Review file · Nature Communications]

Impact of household size and co-resident multimorbidity on unplanned hospitalisation and transition to care home

Corresponding Author: Dr Clare MacRae

Version 0:

Reviewer comments:

Reviewer #1

(Remarks to the Author)

This well-written paper utilises data from a large Welsh cohort study (N~1.5 million people aged 18 years and older) derived by linking census, health and administrative records to explore the impact of household size and of living with someone with multimorbidity on rates of unplanned hospitalisation and transition to living in a care home.

The authors make a clear and compelling case for the need for this study in their introduction and the methods are appropriate and clearly described. Linkage of these data and work to prepare the dataset for analyses will have taken considerable time and effort for which the authors are to be commended.

The findings that living alone or with co-residents with multimorbidity are both associated with slightly elevated rates of the two main outcomes is novel and has potentially important policy implications. This is not only because the population is ageing and the burden of multimorbidity is increasing, which the authors highlight, but also because the proportion of the population who live alone has been rising over recent decades (the latest figures from ONS suggest that 8.4 million people in the UK were living alone in 2023, which represents an 8% increase since 2013)

Major comments

1) The fact that an increasing number of people live alone is one of the reasons that these findings are potentially policy relevant. However, as the number of people living alone has increased, the characteristics of this population has diversified which presents a number of challenges in interpreting these findings. This is not least because our understanding of the health implications of living alone are based on quite outdated literature which assume people living alone are typically older and more vulnerable – while some age groups who live alone may be more vulnerable than their co-residing peers, as a growing number of younger adults also now live alone, whether this is necessarily always the case is less certain. I would therefore suggest that the authors give careful consideration to how best to reflect the potential heterogeneity in one of their main 'exposures' (living alone) as arguably this may be more or less detrimental for some segments of their study population than others.

2) Relating to the point above, the authors should exercise caution in labelling people who comprise 30% of all UK households as 'vulnerable' on the basis that they live alone (as could be implied from the concluding statement of the abstract and discussion).

3) As previous studies have shown that living alone (or being single vs married) is more strongly associated with some adverse age-related health outcomes among men than women, it is good to see that the authors have tested for differences by sex. Did the authors also consider formally testing differences in associations by age? It is noted that continuous and quadratic terms for age were included in the main models to capture the non-linear relationships between age and both outcomes, but there is also a need to consider, that for the reasons outlined above, the strength of associations between the main exposures and outcomes may differ by age, relating to there being different drivers of living alone or with someone with multimorbidity by age and birth cohort.

4) Effect estimates were substantially attenuated after adjustment for covariates. It would be interesting to understand which of the many factors included in the models caused the greatest attenuation – for example, was this largely explained by adjustment for age and sex? Or did other factors also play an important contributing role.

5) As the sample size is very large, even though effect estimates were attenuated considerably after adjustments they still remained statistically significant. However, it would be helpful to see the authors comment on the clinical significance of their findings – for example, how meaningful are HR of 1.06 and 1.03 (as reported in the abstract)? That the effects are very modest does not undermine the importance of the findings and could be more clearly acknowledged.

6) Why was living in a household of 3 or more people selected as the reference group? If it is living alone that is considered to make people 'vulnerable' it may also have been useful to see a comparison of those people living alone vs all those not living alone. My assumption is that this would not show the same effects given Figure 2 shows elevated rates of outcome among those living in 2-person households as well.

Minor comments

1) As the effect estimates presented are hazard ratios it would be more appropriate to refer to rates of unplanned hospitalisation and rates of transition to care home rather than risk.

2) Given a recent paper outlines the case for using the term multiple long-term conditions rather than multimorbidity (see <https://www.bmj.com/content/383/bmj.p2327>) – what is the authors' justification for using the term multimorbidity?

3) Please ensure all authors' names are provided in the same format (e.g. Mizen A, Fry A and Baranyi G)

Reviewer #2

(Remarks to the Author)

The present study examined the association between household size and co-resident multimorbidity with unplanned hospitalization and the transition to living in a care home among adults using census-linked population data in Wales. The topic is interesting, but the methodology appears oversimplified, and the novelty is unclear. It is not surprising that individuals living with residents who have multimorbidity experience worse health outcomes. Existing studies, critiqued by the authors, may be effective because individuals with the most complex health needs are often not cared for at home.

Analyzing the entire adult population in one set of analyses is problematic. The incidence rate of transitioning to living in a care home (which the authors should report instead of risk) was only 1%, likely occurring almost exclusively among older adults. Thus, the findings may not be applicable to the younger population. Similarly, the increased hazard of transitioning to living in a care home associated with living alone, as shown in Figure 2, likely reflects structural differences between individuals living in a two-person household and those living alone (often older, widowed adults). Multivariable adjustment might not be sufficient; propensity score matching or other advanced approaches for dealing with confounding are needed.

When examining the association between household size and the two health outcomes, the authors should additionally stratify individuals living with others by co-resident multimorbidity status and compare each group with those living alone. It is possible that individuals living alone have better health outcomes than those living with multiple persons with multimorbidity.

Given that 47 conditions were considered, the combination of multimorbidities is extensive. No comparisons were made between individuals with different types of multimorbidity.

The authors should report incidence rates instead of risks for the outcomes.

How did the authors determine the list of 47 long-term conditions (data availability, disease prevalence, or another criterion)?

Death is a competing outcome for the two primary outcomes—unplanned hospitalization and transition to a care home. How did the authors address this statistically? The risk of death was higher than the risk of transitioning to living in a care home.

What was the rationale for not accounting for the disease status of co-resident children?

The authors should discuss the limitations of excluding households where more than one resident did not have full health data; this accounted for over 10% of the analytic sample.

Reviewer #3

(Remarks to the Author)

Review of NCOMMS-24-28680

This is a very interesting and innovative study. Using population data from the SAIL databank in Wales, the authors examine how living arrangement and co-morbidity of household members is prospectively associated with two adverse health care outcomes, namely unplanned hospital admission and care home admission. Carefully executed multilevel failure time models demonstrate that those living alone fare worse than those living in larger households, particularly with respect to care home transmissions. Among those living with others, the health condition of household members matters; multimorbidity of housemates means greater chance of adverse outcome. Across models, estimates are attenuated once covariates are adjusted, but the statistically significant, if relatively modest, adjusted estimates suggest that there are important new findings here.

Key strengths of these data include the Census-linked population records which do not require self-report, the use of two outcome variables, and the consideration of multi-morbidity of household members. From my own review of the literature, the latter point is one that has not been widely addressed in extant research. So there is a big innovation in asking the question, and one that has important policy implications.

Here are some limitations that should be addressed in a revision:

1. The abstract is imprecise in parts. Please revisit lines 55-56 and indicate what the "similar associations" are for "both sexes." Does the author wish to say that there is observed effect heterogeneity by sex?
2. One of the problems with the household data is that there is number of members, but who those members are is not specified. Is such information available? It may be informative, for instance, to know whether the person someone resides with is their spouse, their child, or a unrelated roommate. The empirical and policy implications for these various situations may be quite unique.
3. Another analytic issue is the simplification of multi-morbidity as a binary trait. The specific count of comorbid conditions may be important than merely the existence of 2+ conditions; as well, the mix of conditions have quite different implications for caregiving capacity and future risk of adverse outcomes.
4. The Welsh Index of Multiple Deprivation is described as valid for "small areas in Wales" (line 150). Does this imply that the study excluded residents of the large urban areas? Please clarify.
5. In demonstrating the policy significance of this study, it would be helpful to translate some of the adverse outcomes into costs for the health care system.
6. Please indicate whether the study was pre-registered and give other assurances that best practices of open science were followed.

Version 1:

Reviewer comments:

Reviewer #1

(Remarks to the Author)

I thank the authors for carefully considering my previous comments and addressing these satisfactorily in their revisions. I think this paper will make a useful contribution to the literature and I have no further comments.

Reviewer #2

(Remarks to the Author)

The authors have addressed my comments adequately.

Reviewer #3

(Remarks to the Author)

I reviewed the first version of the manuscript, and have now reviewed the revised version. I think the authors have been very attentive to reviewer comments and I have no further comments. I will note that the data have some real limitations--namely, the inability to distinguish co-residents by relationship, but that is a topic for future research.

REVIEWER COMMENTS

Reviewer #1 (Remarks to the Author):

This well-written paper utilises data from a large Welsh cohort study (N~1.5 million people aged 18 years and older) derived by linking census, health and administrative records to explore the impact of household size and of living with someone with multimorbidity on rates of unplanned hospitalisation and transition to living in a care home.

The authors make a clear and compelling case for the need for this study in their introduction and the methods are appropriate and clearly described. Linkage of these data and work to prepare the dataset for analyses will have taken considerable time and effort for which the authors are to be commended.

The findings that living alone or with co-residents with multimorbidity are both associated with slightly elevated rates of the two main outcomes is novel and has potentially important policy implications. This is not only because the population is ageing and the burden of multimorbidity is increasing, which the authors highlight, but also because the proportion of the population who live alone has been rising over recent decades (the latest figures from ONS suggest that 8.4 million people in the UK were living alone in 2023, which represents an 8% increase since 2013)

Major comments

1) The fact that an increasing number of people live alone is one of the reasons that these findings are potentially policy relevant. However, as the number of people living alone has increased, the characteristics of this population has diversified which presents a number of challenges in interpreting these findings. This is not least because our understanding of the health implications of living alone are based on quite outdated literature which assume people living alone are typically older and more vulnerable – while some age groups who live alone may be more vulnerable than their co-residing peers, as a growing number of younger adults also now live alone, whether this is necessarily always the case is less certain. I would therefore suggest that the authors give careful consideration to how best to reflect the potential heterogeneity in one of their main ‘exposures’ (living alone) as arguably this may be more or less detrimental for some segments of their study population than others.

Response 1. We have added further justification for the need to study household composition given that living in single-person households is becoming more common. We have also added sensitivity analyses that examine the effects of household size and co-resident multimorbidity for groups stratified by age into 18-64 and ≥65 years of age.

Please see the Background section on page 3 lines 16-17:

“In contrast to this, living alone can be associated with isolation and increased use of unplanned hospital care,⁸ **and it is a growing concern as it becomes more common. For example, the number of individuals living alone in the UK increased by 8% between 2013 and 2023.**”⁹”

Please see the Methods section on page 5 lines 25-29:

“First, to examine how the effect of household size and co-resident multimorbidity might vary for particular segments of the population, given that these exposures might affect people differently depending on their age, subgroup analyses were performed in individuals aged 18-64 and in those aged 65 and older.”

Please see the Results section, page 9 on lines 6-11:

“Living with coresident(s) with versus without multimorbidity was associated with a slightly higher aHR of unplanned hospitalisation in three-or-more-person households in the younger and in two-person households for the older subgroup (Supplementary Table S11). In models examining the transition to living in a care home, there were larger differences in aHR between those who did and did not live with co-residents with multimorbidity in the younger versus the older subgroup (Supplementary Table S12).”

Relating to the point above, the authors should exercise caution in labelling people who comprise 30% of all UK households as ‘vulnerable’ on the basis that they live alone (as could be implied from the concluding statement of the abstract and discussion).

Response 2. We have amended this language.

Please see the Abstract on page 2 lines 22-23:

“Understanding the mechanisms behind these associations is needed to inform targeted intervention strategies.”

Please see the Conclusion on page 12 line 25:

“Therefore, focused research and public health attention is needed to identify individuals and households who would benefit from targeted support.”

3) As previous studies have shown that living alone (or being single vs married) is more strongly associated with some adverse age-related health outcomes among men than women, it is good to see that the authors have tested for differences by sex. Did the authors also consider formally testing differences in associations by age? It is noted that continuous and quadratic terms for age were included in the main models to capture the non-linear relationships between age and both outcomes, but there is also a need to consider, that for the reasons outlined above, the strength of associations between the main exposures and outcomes may differ by age, relating to there being different drivers of living alone or with someone with multimorbidity by age and birth cohort.

Response 3. We have included subgroup analyses for those aged 65 years and younger and those aged over 65 years in subgroup analyses 1 and 2 in Supplementary Tables 11 and 12 to test the associations between household size

and each outcome for different age groups. We have also added a further research recommendation to understand the reasons for living alone and how the effects of household composition might be different by age cohort.

Please see Response 1.

Please see the Discussion section on page 12 lines 12-16:

“Using more granular datasets that provide an understanding of the reasons for living alone by life stage and the mechanisms by which living alone or with co-residents with multimorbidity in different ages and birth cohorts increases the event rate of unplanned hospitalisation and transition to living in a care home could be helpful precursors for the development and economic evaluation of complex interventions.”

4) Effect estimates were substantially attenuated after adjustment for covariates. It would be interesting to understand which of the many factors included in the models caused the greatest attenuation – for example, was this largely explained by adjustment for age and sex? Or did other factors also play an important contributing role.

Response 4. We have incorporated coefficients for unadjusted, partially adjusted, and fully adjusted models into the results section to enhance the visibility of the importance of confounder adjustment.

Please see the Methods section on page 6 lines 15-17:

“Model coefficients were presented for unadjusted, partially adjusted (incorporating age group, sex, and socioeconomic position), and fully adjusted models.”

Please see the Results section on page 8 lines 1-5:

“Effect sizes and the differences in hazard ratios between living arrangements were the largest in the unadjusted models. The greatest degree of attenuation was seen when age, sex, and socioeconomic deprivation were incorporated (in the partially adjusted models), and a smaller degree of attenuation was seen with the addition of the number of long-term conditions, body mass index, and smoking in the fully adjusted models (Table 3, Figure 2, Supplementary Table S9).”

Please see the Results section on page 8 lines 19-25:

“Both before and after partial and full adjustment for demographic, health, and health-related behaviours, the difference between living alone and the highest-risk group of the co-habiting living arrangements was considerable and was larger than seen in the unplanned hospitalisation models. However, a similar degree of attenuation to the unplanned hospitalisation models was seen in the magnitude of effect sizes and the difference in hazards between living arrangements was

seen when sociodemographic characteristics were included in the partially adjusted model and health behaviours in the fully adjusted model (Supplementary Table S10)."

Please see Table 3.

Please see Supplementary Tables S9 and S10.

5) As the sample size is very large, even though effect estimates were attenuated considerably after adjustments they still remained statistically significant. However, it would be helpful to see the authors comment on the clinical significance of their findings – for example, how meaningful are HR of 1.06 and 1.03 (as reported in the abstract)? That the effects are very modest does not undermine the importance of the findings and could be more clearly acknowledged.

Response 5. We have added additional text to emphasise the clinical importance of the findings.

Please see the Conclusion on page 12 line 20:

"Our study finds that living alone and living with co-residents with multimorbidity is independently associated with the transition to living in a care home and to a smaller extent with unplanned hospitalisation which is clinically important given that it is a common event."

6) Why was living in a household of 3 or more people selected as the reference group? If it is living alone that is considered to make people 'vulnerable' it may also have been useful to see a comparison of those people living alone vs all those not living alone. My assumption is that this would not show the same effects given Figure 2 shows elevated rates of outcome among those living in 2-person households as well.

Response 6. To improve the visibility of the hazard ratios across different household compositions, including living alone, we have amended the exposure of interest to incorporate every level of household composition. We have made "Living alone" the reference point for the models.

Please see the Abstract Results section on page 2 lines 14-18.

"In three-or-more-person households, compared to living alone, the adjusted hazard ratio (aHR) for unplanned hospitalisation was 0.87 (95%CI 0.86-0.88) when co-residents did not have multimorbidity versus 0.92 (95%CI 0.91-0.93) when they did. Differences were more substantial for care home transition (co-resident without multimorbidity aHR 0.57, 95% CI 0.55-0.59; co-resident with multimorbidity aHR 0.78, 95% CI 0.75-0.80)."

Please see the Methods section on page 5 lines 20-24:

“To do this, we created an exposure variable of interest by categorising each participant into one of five groups according to their living arrangement: living alone, two-person household where the co-resident had multimorbidity, two-person household where the co-resident did not have multimorbidity, three-or-more-person household where one or more co-residents had multimorbidity, and three-or-more-person household where no co-residents had multimorbidity.”

Please see the Results section on page 8 lines 5-13:

“For example, in those living in three-or-more-person households, compared to living alone, the unadjusted HR of unplanned hospitalisation when no co-residents had multimorbidity was lowest (aHR 0.37, 95%CI 0.37-0.37) and was intermediate when at least one co-resident had multimorbidity (aHR 0.50, 95%CI 0.50-0.51). The same comparison in the fully adjusted models showed a similar pattern but with weaker associations: aHR 0.87 (95%CI 0.86-0.88) compared to aHR 0.92 (95%CI 0.91-0.93). This effect was restricted to people living in three-or-more-person households, as there was little difference in aHR for those living in two-person households, regardless of whether their co-resident had multimorbidity (0.91, 95%CI 0.90-0.92) or not (0.89, 95%CI 0.89-0.90).”

Please see the Results section on page 8 lines 26-32:

“In people living in three-or-more-person households, compared to living alone, the fully adjusted aHR of transition to care home was 0.57 (95% CI 0.55-0.59) when no co-residents had multimorbidity which was substantially lower than when one or more co-residents had multimorbidity (0.78, 95%CI 0.75-0.80) (Table 3). The same comparison of the difference in aHRs in two-person households was smaller but still moderate in size. Living with a co-resident who did not have multimorbidity was associated with the lowest hazard (aHR 0.71, 95% CI 0.69-0.72) and there was a significantly higher hazard for those living with a co-resident who did have multimorbidity (0.78, 95%CI 0.77-0.79).”

Please see Table 3.

Please see Figure 2.

Minor comments

1) As the effect estimates presented are hazard ratios it would be more appropriate to refer to rates of unplanned hospitalisation and rates of transition to care home rather than risk.

Response 7. We have included rates per 1000 person-years for each of the levels of each covariate and changed the description from “risk” to “hazard” throughout the manuscript.

Please see the Results section on page 7 lines 29-32:

“People living alone had the highest rate of unplanned hospitalisation (91.96/1000 person-years), with lower rates in all other household arrangements (ranging from 39.88/1000 person-years in three-person households where co-residents did not have multimorbidity to 85.46/1000 person-years in two-person households where the co-resident did have multimorbidity) (Table 3).”

Please see the Results section on page 8 lines 15-18:

“The rate of transitioning to live in a care home was also highest in people who lived alone (5.88/1000 person-years) and was much lower in those living in other household arrangements (from 0.14 in three-or-more-person households with a co-resident who did not have multimorbidity to 2.82 in two-person households where the co-resident did have multimorbidity) (Table 3).”

Please see Table 2.

Please see Supplementary Tables S9 and S10.

2) Given a recent paper outlines the case for using the term multiple long-term conditions rather than multimorbidity (see <https://www.bmj.com/content/383/bmj.p2327>) – what is the authors’ justification for using the term multimorbidity?

Response 8. Although ‘multiple long-term conditions’ has been proposed as a better term than multimorbidity in the UK, this is not true internationally (and we note that the current call for papers by Nature Journals uses the term multimorbidity). We, therefore, haven’t changed this, but can amend if “multiple long-term conditions” is preferred.

3) Please ensure all authors’ names are provided in the same format (e.g. Mizen A, Fry A and Baranyi G).

Response 9. We apologise for this oversight.

Please see the amended author list on Page 1:

Reviewer #2 (Remarks to the Author):

The present study examined the association between household size and co-resident multimorbidity with unplanned hospitalization and the transition to living in a care home among adults using census-linked population data in Wales. The topic is interesting, but the methodology appears oversimplified, and the novelty is unclear. It is not surprising that individuals living with residents who have multimorbidity experience worse health outcomes. Existing studies, critiqued by the authors, may be effective because individuals with the most complex health needs are often not cared for at home.

Analyzing the entire adult population in one set of analyses is problematic. The incidence rate of transitioning to living in a care home (which the authors should report instead of risk) was only 1%, likely occurring almost exclusively among older adults. Thus, the findings may not be applicable to the younger population. Similarly, the increased hazard of transitioning to living in a care home associated with living alone, as shown in Figure 2, likely reflects structural differences between individuals living in a two-person household and those living alone (often older, widowed adults).

Response 10. Thank you for making this important point about ensuring that the study reports the rate of the outcome and the need to report rates and likelihoods of the outcomes for different age groups within the study population. Your recommendations have shaped our responses documented above where Reviewer 1 has made similar points. Please see our responses above to Reviewer 1 as also responding to your comments. Specifically, please see Response 1 and Response 3, where we have provided subgroup analyses for the populations stratified by age, and Response 7 where we have provided rates per 1000 person-years for every level of every covariate and changed the wording throughout the manuscript.

Multivariable adjustment might not be sufficient; propensity score matching or other advanced approaches for dealing with confounding are needed.

Response 11. Propensity score matching is complicated in this setting as we have multiple non-binary exposure variables (e.g., household size and co-resident multimorbidity). In this setting, the propensity score model would require a multinomial model with the propensity calculated for different categories. Following feedback, we have adjusted the analysis to account for competing risks, which further complicates the analysis for advanced methods such as G-estimation to account for unmeasured confounding. For this reason, we are unclear regarding the implementation and/or added benefits of advanced adjustments over and above multivariable adjustment.

When examining the association between household size and the two health outcomes, the authors should additionally stratify individuals living with others by co-resident multimorbidity status and compare each group with those living alone. It is possible that individuals living alone have better health outcomes than those living with multiple persons with multimorbidity.

Response 12. Please see Response 6 where we have adopted “Living alone” as the reference point for all the models.

Response 13. We have included a new sensitivity analysis to examine the effect of household composition using a covariate that stratifies co-resident morbidities into a more granular set of exposures,

Please see the Methods section on page 6 lines 3-5:

“a more granular examination of co-resident morbidity was assessed by disaggregating the exposure variable of interest, replacing multimorbidity as a binary variable with zero to one, two to three, or four or more long-term conditions.”

Please see the Results section on Page 9 lines 18-20:

“there was no difference in the hazard of unplanned hospitalisation or transition to living in a care home between different numbers of co-resident long-term conditions.”

Please see the results of this analysis in Supplementary Table S16.

Given that 47 conditions were considered, the combination of multimorbidities is extensive. No comparisons were made between individuals with different types of multimorbidity.

Response 14. We have included the examination of household size and co-resident mental-physical multimorbidity within sensitivity analyses.

Please see the Methods section on page 5 line 32 and page 6 lines 1-3:

“Third, household size and co-resident mental-physical multimorbidity defined as one or more physical health and one or more mental health long-term conditions was tested to examine whether the same or different results were found using an alternative definition of multimorbidity.¹⁶”

Please see the Results section on page 9 lines 16-18:

“There were no substantial differences in the hazard of unplanned hospitalisation or transition to living in a care home when defining co-resident multimorbidity as the coexistence of both mental and physical long-term conditions.”

Please see the results of this analysis in Supplementary Table S15.

The authors should report incidence rates instead of risks for the outcomes.

Response 15. Please see Response 7 (and amended Table 3 and Supplementary Tables S9 and S10) where we have provided rates per 1000 person-years for every level of every covariate. We have changed the description from “risk” to “hazard” throughout the manuscript.

How did the authors determine the list of 47 long-term conditions (data availability, disease prevalence, or another criterion)?

Response 16. The choice was based on an international consensus recommendation for conditions to include in multimorbidity studies and a quantitative examination of multimorbidity prevalence estimates when comparing different numbers and selections of long-term conditions in the measurement of multimorbidity. We have made this point clearer in the Methods section.

Please see the Methods section on page 4 lines 22-26:

“Methods used to select the constituent conditions (**based on a recent Delphi consensus study on the measurement of multimorbidity¹⁷ and a quantitative examination of the implications of considering different numbers and selections of long-term conditions in the count of multimorbidity¹⁸**) and to define each condition according to each data source¹⁹ are described in Supplementary Information Panel S1.”

Death is a competing outcome for the two primary outcomes—unplanned hospitalization and transition to a care home. How did the authors address this statistically? The risk of death was higher than the risk of transitioning to living in a care home.

Response 17. Thank you for making this very important point. When the analysis was designed there was no straightforward way to account for both clustering by household and competing mortality risk in a survival model, and we chose to prioritise clustering by household given that households were the focus of the study. We have sought additional statistical input and have now applied a recently developed method of incorporating the competing risk of death into models that can also account for household random effects within a survival analysis.

Please see the Abstract Methods section on page 2 lines 9-10:

“We examined how household size and co-resident multimorbidity were associated with 5-year hazard of two outcomes (unplanned hospitalisation and care home transition) **using multilevel multistate models accounting for both competing risk of death and clustering within households.**”

Please see the Methods section on page 6 lines 6-14:

“**Multistate Cox proportional hazard models were used to examine the hazard ratio of each outcome using separate models. Coefficients for transition one (transitioning from home to first unplanned hospitalisation or home to transition to living in a care home, for separate models) were reported in the results section. The remaining two transitions, from home to death (therefore allowing the model to incorporate the competing risk of death), and transition three which was unplanned hospitalisation or care home to death, were not reported. No violations of the proportional hazard assumption were found on visual inspection for deviation from the zero slope of plotted Schoenfeld residuals.** The outcome was measured using the appropriate data structure by incorporating household-level random effects i.e., **to account for clustering of effects within household units.**”

What was the rationale for not accounting for the disease status of co-resident children?

Response 18. We have added further clarification regarding this methodological choice to the manuscript.

Please see the Methods section on page 4 lines 15-17:

“Child co-residents were not included in counts of co-resident multimorbidity status, given that the study aimed to ascertain the effect of differing levels of potential care available from household co-residents.”

The authors should discuss the limitations of excluding households where more than one resident did not have full health data; this accounted for over 10% of the analytic sample.

Response 19. We have added to the limitations section of the manuscript.

Please see the Discussion section on page 10 lines 11-15:

“A total of 11.5% of households were excluded because one or more residents did not have linked primary care data. However, incorporating primary care data considerably improves the robustness of multimorbidity measurement²² and was therefore prioritised in the study design over including all households but only ascertaining morbidity using hospital inpatient data.”

Reviewer #3 (Remarks to the Author):

Review of NCOMMS-24-28680

This is a very interesting and innovative study. Using population data from the SAIL databank in Wales, the authors examine how living arrangement and co-morbidity of household members is prospectively associated with two adverse health care outcomes, namely unplanned hospital admission and care home admission.

Carefully executed multilevel failure time models demonstrate that those living alone fare worse than those living in larger households, particularly with respect to care home transmissions. Among those living with others, the health condition of household members matters; multi-morbidity of housemates means greater chance of adverse outcome. Across models, estimates are attenuated once covariates are adjusted, but the statistically significant, if relatively modest, adjusted estimates suggest that there are important new findings here.

Key strengths of these data include the Census-linked population records which do not require self-report, the use of two outcome variables, and the consideration of multi-morbidity of household members. From my own review of the literature, the latter point is one that has not been widely addressed in extant research. So there is a big innovation in asking the question, and one that has important policy implications.

Here are some limitations that should be addressed in a revision:

1. The abstract is imprecise in parts. Please revisit lines 55-56 and indicate what the “similar associations” are for “both sexes.” Does the author wish to say that there is observed effect heterogeneity by sex?

Response 20. We have revised the manuscript abstract to reflect the changes to the study design and to improve clarity.

Please see the Abstract Results section on page 2 lines 12-18:

“The highest rates of unplanned hospitalisation and care home transition were in those living alone (92.0 and 5.9 events/1000py respectively). Event rates were lower in all shared households, and lowest when co-residents did not have multimorbidity. In three-or-more-person households, compared to living alone, the adjusted hazard ratio (aHR) for unplanned hospitalisation was 0.87 (95%CI 0.86-0.88) when co-residents did not have multimorbidity versus 0.92 (95%CI 0.91-0.93) when they did. Differences were more substantial for care home transition (co-resident without multimorbidity aHR 0.57, 95% CI 0.55-0.59; co-resident with multimorbidity aHR 0.78, 95% CI 0.75-0.80).”

One of the problems with the household data is that there is number of members, but who those members are is not specified. Is such information available? It may be informative, for instance, to know whether the person someone resides with is their spouse, their child, or a unrelated roommate. The empirical and policy implications for these various situations may be quite unique.

Response 21. Unfortunately, kinship and care relationships within households was not available within these routinely collected data. We have added further clarification to the main manuscript.

Please see the Methods section within strengths and limitations on page 10 lines 15-18:

“Potential confounders of the associations for which we were unable to adjust include the provision of both formal and informal home care (including from family or friends who were not co-residents), or how co-residents are related, which are likely to be important factors were not possible to ascertain within these data.”

Please see the Discussion section on page 12 on line 11:

“Future research would ideally incorporate measurements of accumulated individual and household exposures over the life course to understand how physical and mental health, material deprivation, household social dynamics including household kinship and care relationships, and health and care services might mutually reinforce each other over time.”

3. Another analytic issue is the simplification of multi-morbidity as a binary trait. The specific count of comorbid conditions may be important than merely the existence of 2+ conditions; as well, the mix of conditions have quite different implications for caregiving capacity and future risk of adverse outcomes.

Response 22. Please see Response 13 where sensitivity analyses have been added to examine the effect of co-resident number of long-term conditions with both outcomes.

Please see Response 14 where sensitivity analyses examine co-resident mental-physical multimorbidity as an exposure.

4. The Welsh Index of Multiple Deprivation is described as valid for “small areas in Wales” (line 150). Does this imply that the study excluded residents of the large urban areas? Please clarify.

Response 23. The Welsh Index of Multiple Deprivation is an allocation of deprivation status to geographical areas called “Lower Super Output Areas” or LSOAs. These are statistical geographical areas consisting of 400 to 1,200 households each and cover all of Wales.

Please see the Methods section on page 5 lines 13-14:

“Multiple Deprivation, the Welsh Government’s official measure of relative deprivation²⁰ **for every Lower Super Output Area (geographies comprising between 400 and 1,200 households) in Wales.**²¹”

5. In demonstrating the policy significance of this study, it would be helpful to translate some of the adverse outcomes into costs for the health care system.

Response 24. Thank you for making this point. We agree that an important extension of this work is to understand the cost implications of different household compositions for health and care systems in Wales. This work requires complex methodology and specialist input and therefore we have added this to recommendations for future research.

Please see the Discussion section on page 12 line 12-16:

“Using more granular datasets that provide an understanding of the reasons for living alone by life stage and the mechanisms by which living alone or with co-residents with multimorbidity in different ages and birth cohorts increases the event rate of unplanned hospitalisation and transition to living in a care home could be helpful precursors for the development **and economic evaluation** of complex interventions.”

6. Please indicate whether the study was pre-registered and give other assurances that best practices of open science were followed.

Response 25. Our study has been approved by the SAIL Databank Independent Governance Review Panel (project number 1350) – details of this panel can be found at <https://saildatabank.com/governance/approvals-public-engagement/information-governance/>. Additionally, given that we had access to data made available by the Office for National Statistics (ONS) from the 2011 Wales Census, our project was additionally reviewed by the ONS Research Accreditation Panel (project number 2022/139). Both applications involved the submission of a research protocol. All research outputs were manually screened by members of the SAIL Databank staff who cross-examined outputs against the peer-reviewed study protocol prior to release from the secure research environment.

Please see the Methods section on page 6 lines 25-26:

“All research outputs were manually screened and cross-checked against the approved research protocol before being released from the secure research environment.”